# A Communication-Efficient Parallel Algorithm for Decision Tree

**Qi Meng**[1,*]**, Guolin Ke**[2,*]**, Taifeng Wang**[2]**, Wei Chen**[2]**, Qiwei Ye**[2]**,**
**Zhi-Ming Ma**[3]**, Tie-Yan Liu**[2]
[1]Peking University    [2]Microsoft Research
[3]Chinese Academy of Mathematics and Systems Science
[1]qimeng13@pku.edu.cn; [2]{Guolin.Ke, taifengw, wche, qiwye, tie-yan.liu}@microsoft.com;
[3]mazm@amt.ac.cn

## Abstract

Decision tree (and its extensions such as Gradient Boosting Decision Trees and Random Forest) is a widely used machine learning algorithm, due to its practical effectiveness and model interpretability. With the emergence of big data, there is an increasing need to parallelize the training process of decision tree. However, most existing attempts along this line suffer from high communication costs. In this paper, we propose a new algorithm, called *Parallel Voting Decision Tree (PV-Tree)*, to tackle this challenge. After partitioning the training data onto a number of (e.g., $M$) machines, this algorithm performs both local voting and global voting in each iteration. For local voting, the top-$k$ attributes are selected from each machine according to its local data. Then, globally top-$2k$ attributes are determined by a majority voting among these local candidates. Finally, the full-grained histograms of the globally top-$2k$ attributes are collected from local machines in order to identify the best (most informative) attribute and its split point. PV-Tree can achieve a very low communication cost (independent of the total number of attributes) and thus can scale out very well. Furthermore, theoretical analysis shows that this algorithm can learn a near optimal decision tree, since it can find the best attribute with a large probability. Our experiments on real-world datasets show that PV-Tree significantly outperforms the existing parallel decision tree algorithms in the trade-off between accuracy and efficiency.

## 1 Introduction

Decision tree [16] is a widely used machine learning algorithm, since it is practically effective and the rules it learns are simple and interpretable. Based on decision tree, people have developed other algorithms such as Random Forest (RF) [3] and Gradient Boosting Decision Trees (GBDT) [7], which have demonstrated very promising performances in various learning tasks [5].

In recent years, with the emergence of very big training data (which cannot be held in one single machine), there has been an increasing need of parallelizing the training process of decision tree. To this end, there have been two major categories of attempts: [2].

---

[*]Denotes equal contribution. This work was done when the first author was visiting Microsoft Research Asia.
[2]There is another category of works that parallelize the tasks of sub-tree training once a node is split [15], which require the training data to be moved from machine to machine for many times and are thus inefficient. Moreover, there are also some other works accelerating decision tree construction by using pre-sorting [13] [19] [11] and binning [17] [8] [10], or employing a shared-memory-processors approach [12] [1]. However, they are out of our scope.

*Attribute-parallel*: Training data are vertically partitioned according to the attributes and allocated to different machines, and then in each iteration, the machines work on non-overlapping sets of attributes in parallel in order to find the best attribute and its split point (suppose this best attribute locates at the $i$-th machine) [19] [11] [20]. This process is communicationally very efficient. However, after that, the re-partition of the data on other machines than the $i$-th machine will induce very high communication costs (proportional to the number of data samples). This is because those machines have no information about the best attribute at all, and in order to fulfill the re-partitioning, they must retrieve the partition information of every data sample from the $i$-th machine. Furthermore, as each worker still has full sample set, the partition process is not parallelized, which slows down the algorithm.

*Data-parallel*: Training data are horizontally partitioned according to the samples and allocated to different machines. Then the machines communicate with each other the local histograms of all attributes (according to their own data samples) in order to obtain the global attribute distributions and identify the best attribute and split point [12] [14]. It is clear that the corresponding communication cost is very high and proportional to the total number of attributes and histogram size. To reduce the cost, in [2] and [21] [10], it was proposed to exchange quantized histograms between machines when estimating the global attribute distributions. However, this does not really solve the problem – the communication cost is still proportional to the total number of attributes, not to mentioned that the quantization may hurt the accuracy.

In this paper, we proposed a new data-parallel algorithm for decision tree, called *Parallel Voting Decision Tree (PV-Tree)*, which can achieve much better balance between communication efficiency and accuracy. The key difference between conventional data-parallel decision tree algorithm and PV-Tree lies in that the former only trusts the globally aggregated histogram information, while the latter leverages the local statistical information contained in each machine through a two-stage voting process, thus can significantly reduce the communication cost. Specifically, PV-Tree contains the following steps in each iteration. 1) *Local voting*. On each machine, we select the top-$k$ attributes based on its local data according to the informativeness scores (e.g., risk reduction for regression, and information gain for classification). 2) *Global voting*. We determine global top-$2k$ attributes by a majority voting among the local candidates selected in the previous step. That is, we rank the attributes according to the number of local machines who select them, and choose the top $2k$ attributes from the ranked list. 3) *Best attribute identification*. We collect the full-grained histograms of the globally top-$2k$ attributes from local machines in order to compute their global distributions. Then we identify the best attribute and its split point according to the informativeness scores calculated from the global distributions.

It is easy to see that PV-Tree algorithm has a very low communication cost. It does not need to communicate the information of all attributes, instead, it only communicates indices of the locally top-$k$ attributes per machine and the histograms of the globally top-$2k$ attributes. In other words, its communication cost is independent of the total number of attributes. This makes PV-Tree highly scalable. On the other hand, it can be proven that PV-Tree can find the best attribute with a large probability, and the probability will approach 1 regardless of $k$ when the training data become sufficiently large. In contrast, the data-parallel algorithm based on quantized histogram could fail in finding the best attribute, since the bias introduced by histogram quantization cannot be reduced to zero even if the training data are sufficiently large.

We have conducted experiments on real-world datasets to evaluate the performance of PV-Tree. The experimental results show that PV-Tree has consistently higher accuracy and training speed than all the baselines we implemented. We further conducted experiments to evaluate the performance of PV-Tree in different settings (e.g., with different numbers of machines, different values of $k$). The experimental results are in accordance with our theoretical analysis.

## 2  Decision Tree

Suppose the training data set $D_n = \{(x_{i,j}, y_i); i = 1, \cdots, n, j = 1, \cdots, d\}$ are independently sampled from $\prod_{j=1}^d \mathcal{X}_j \times \mathcal{Y}$ according to $(\prod_{j=1}^d P_{X_j})P_{Y|X}$. The goal is to learn a regression or classification model $f \in \mathcal{F} : \prod_{j=1}^d \mathcal{X}_j \to \mathcal{Y}$ by minimizing loss functions on the training data, which hopefully could achieve accurate prediction for the unseen test data.

Decision tree[16, 18] is a widely used model for both regression [4] and classification [18]. A typical decision tree algorithm is described in Alg 1. As can be seen, the tree growth procedure is recursive, and the nodes will not stop growing until they reach the *stopping criteria*. There are two important functions in the algorithm: *FindBestSplit* returns the best split point *{attribute, threshold}* of a node, and *Split* splits the training data according to the best split point. The details of *FindBestSplit* is given in Alg 2: first histograms of the attributes are constructed (for continuous attributes, one usually converts their numerical values to finite bins for ease of compuation) by going over all training data on the current node; then all *bins* (split points) are traversed from left to right, and *leftSum* and *rightSum* are used to accumulate sum of left and right parts of the split point respectively. When selecting the best split point, an informativeness measure is adopted. The widely used informative measures are *information gain* and *variance gain* for classification and regression, respectively.

---

**Algorithm 1** BulidTree

**Input:** Node N, Dateset D
**if** StoppingCirteria(D) **then**
    N.output = Prediction(D)
**else**
    bestSplit = FindBestSplit(D)
    (DL, DR) = Split(D, N, bestSplit)
    BuildTree(N.leftChild, DL)
    BuildTree(N.rightChild, DR)
**end if**

---

**Definition 2.1** *[6][16] In classification, the information gain (IG) for attribute $X_j \in [w_1, w_2]$ at node O, is defined as the entropy reduction of the output $Y$ after splitting node O by attribute $X_j$ at w, i.e.,*

$$IG_j(w; O) = \mathcal{H}_j - (\mathcal{H}_j^l(w) + \mathcal{H}_j^r(w))$$
$$= P(w_1 \le X_j \le w_2)H(Y|w_1 \le X_j \le w_2) - P(w_1 \le X_j < w)H(Y|w_1 \le X_j < w)$$
$$- P(w \le X_j \le w_2)H(Y|w \le X_j \le w_2),$$

*where $H(\cdot|\cdot)$ denotes the conditional entropy.*

*In regression, the variance gain (VG) for attribute $X_j \in [w_1, w_2]$ at node O, is defined as variance reduction of the output $Y$ after splitting node O by attribute $X_j$ at w, i.e.,*

$$VG_j(w; O) = \sigma_j - (\sigma_j^l(w) + \sigma_j^r(w))$$
$$= P(w_1 \le X_j \le w_2)Var[Y|w_1 \le X_j \le w_2] - P(w_1 \le X_j < w)Var[Y|w_1 \le X_j < w]$$
$$- P(w_2 \ge X_j \ge w)Var[Y|w_2 \ge X_j \ge w],$$

*where $Var[\cdot|\cdot]$ denotes the conditional variance.*

# 3 PV-Tree

In this section, we describe our proposed PV-Tree algorithm for parallel decision tree learning, which has a very low communication cost, and can achieve a good trade-off between communication efficiency and learning accuracy.

PV-Tree is a data-parallel algorithm, which also partitions the training data onto $M$ machines just like in [2] [21]. However, its design principal is very different. In [2][21], one does not trust the local information about the attributes in each machine, and decides the best attribute and split point only based on the aggregated global histograms of the attributes. In contrast, in PV-Tree, we leverage the meaningful statistical information about the attributes contained in each local machine, and make decisions through a two-stage (local and then global) voting process. In this way, we can significantly reduce the communication cost since we do not need to communicate the histogram information of all the attributes across machines, instead, only the histograms of those attributes that survive in the voting process.

The flow of PV-tree algorithm is very similar to the standard decision tree, except function *FindBestSplit*. So we only give the new implementation of this function in Alg 3, which contains following three steps:

*Local Voting*: We select the top-$k$ attributes for each machine based on its local data set (according to the informativeness scores, e.g., information gain for classification and variance reduction for regression), and then exchange indices of the selected attributes among machines. Please note that the communication cost for this step is very low, because only the indices for a small number of (i.e., $k \times M$) attributes need to be communicated.

*Global Voting*: We determine the globally top-$2k$ attributes by a majority voting among all locally selected attributes in the previous step. That is, we rank the attributes according to the number of

local machines who select them, and choose the top-$2k$ attributes from the ranked list. It can be proven that when the local data are big enough to be statistically representative, there is a very high probability that the top-$2k$ attributes obtained by this majority voting will contain the globally best attribute. Please note that this step does not induce any communication cost.

*Best Attribute Identification*: We collect full-grained histograms of the globally top-$2k$ attributes from local machines in order to compute their global distributions. Then we identify the best attribute and its split point according to the informativeness scores calculated from the global distributions. Please note that the communication cost for this step is also low, because we only need to communicate the histograms of $2k$ pre-selected attributes (but not all attributes).[3] As a result, PV-Tree algorithm can scale very well since its communication cost is independent of both the total number of attributes and the total number of samples in the dataset.

In next section, we will provide theoretical analysis on accuracy guarantee of PV-Tree algorithm.

---

**Algorithm 2** FindBestSplit

**Input:** DataSet D
**for all** X in D.Attribute **do**
  ▷ *Construct Histogram*
  H = new Histogram()
  **for all** x in X **do**
    H.binAt(x.bin).Put(x.label)
  **end for**
  ▷ *Find Best Split*
  leftSum = new HistogramSum()
  **for all** bin in H **do**
    leftSum = leftSum + H.binAt(bin)
    rightSum = H.AllSum - leftSum
    split.gain = CalSplitGain(leftSum, rightSum)
    bestSplit = ChoiceBetterOne(split,bestSplit)
  **end for**
**end for**
**return** bestSplit

---

**Algorithm 3** PV-Tree_FindBestSplit

**Input:** Dataset D
localHistograms = ConstructHistograms(D)
▷ *Local Voting*
splits = []
**for all** H in localHistograms **do**
  splits.Push(H.FindBestSplit())
**end for**
localTop = splits.TopKByGain(K)
▷ *Gather all candidates*
allCandidates = AllGather(localTop)
▷ *Global Voting*
globalTop = allCandidates.TopKByMajority(2*K)
▷ *Merge global histograms*
globalHistograms = Gather(globalTop, localHistograms)
bestSplit = globalHistograms.FindBestSplit()
**return** bestSplit

---

## 4 Theoretical Analysis

In this section, we conduct theoretical analysis on proposed PV-Tree algorithm. Specifically, we prove that, PV-Tree can select the best (most informative) attribute in a large probability, for both classification and regression. In order to better present the theorem, we firstly introduce some notations[4] In classification, we denote $IG_j = \max_w IG_j(w)$, and rank $\{IG_j; j \in [d]\}$ from large to small as $\{IG_{(1)}, ..., IG_{(d)}\}$. We call the attribute $j_{(1)}$ the most informative attribute. Then, we denote $l_{(j)}(k) = \frac{|IG_{(1)} - IG_{(j)}|}{2}, \forall j \geq k+1$ to indicate the distance between the largest and the $k$-th largest IG. In regression, $l_{(j)}(k)$ is defined in the same way, except replacing IG with VG.

**Theorem 4.1** *Suppose we have $M$ local machines, and each one has $n$ training data. PV-Tree at an arbitrary tree node with local voting size $k$ and global majority voting size $2k$ will select the most informative attribute with a probability at least*

$$\sum_{m=[M/2+1]}^{M} C_M^m \left( 1 - \left( \sum_{j=k+1}^{d} \delta_{(j)}(n,k) \right) \right)^m \left( \sum_{j=k+1}^{d} \delta_{(j)}(n,k) \right)^{M-m},$$

*where $\delta_{(j)}(n,k) = \alpha_{(j)}(n) + 4e^{-c_{(j)}n\left(l_{(j)}(k)\right)^2}$ with $\lim_{n \to \infty} \alpha_{(j)}(n) = 0$ and $c_{(j)}$ is constant.*

Due to space restrictions, we briefly illustrate the proof idea here and leave detailed proof to supplementary materials. Our proof contains two parts. (1) For local voting, we find a sufficient condition to guarantee a similar rank of attributes ordered by information gain computed based on local data and full data. Then, we derive a lower bound of probability to make the sufficient condition holds by

using concentration inequalities. (2) For global voting, we select top-$2k$ attributes. It's easy to proof that we can select the most informative attribute if only no less than $[M/2+1]$ of all machines select it.[5] Therefore, we can calculate the probability in the theorem using binomial distribution.

Regarding Theorem 4.1, we have following discussions on factors that impact the lower bound for probability of selecting the best attribute.

1.*Size of local training data $n$:* Since $\delta_{(j)}(n,k)$ decreased with $n$, with more and more local training data, the lower bound will increase. That means, if we have sufficiently large data, PV-Tree will select the best attribute with almost probability 1.

2. *Input dimension $d$:* It is clear that for fixed local voting size $k$ and global voting size $2k$, with $d$ increasing, the lower bound is decreasing. Consider the case that the number of attributes become 100 times larger. Then the terms in the summation (from $\sum_{j=k+1}^{d}$ to $\sum_{j=k+1}^{100d}$) is roughly 100 times larger for a relatively small $k$. But there must be many attributes away from attribute (1) and $l_{(j)}(k)$ is a large number which results in a small $\delta_{(j)}(n,k)$. Thus we can say that the bound in the theorem is not sensitive with $d$.

3. *Number of machines $M$:* We assume the whole training data size $N$ is fixed and the local data size $n = \frac{N}{M}$. Then on one hand, as $M$ increases, $n$ decreases, and therefore the lower bound will decrease due to larger $\delta_j(n,k)$. On the other hand, because function $\sum_{m=[M/2+1]}^{M} C_M^m p^m (1-p)^{M-m}$ will approach 1 as $M$ increases when $p > 0.5$ [[23]], the lower bound will increase. In other words, the number of machines $M$ has dual effect on the lower bound: with more machines, local data size becomes smaller which reduces the accuracy of local voting, however, it also leads to more copies of local votes and thus increase the reliability of global voting. Therefore, in terms of accuracy, there should be an optimal number of machines given a fixed-size training data.[6]

4. *Local/Global voting size $k/2k$:* Local/Global voting size $k/2k$ influence $l_{(j)}(k)$ and the terms in the summation in the lower bound . As $k$ increases, $l_{(j)}(k)$ increases and the terms in the summation decreases, and the lower bound increases. But increasing $k$ will bring more communication and calculating time. Therefore, we should better select a moderate $k$. For some distributions, especially for the distributions over high-dimensional space, $l_{(j)}(k)$ is less sensitive to $k$, then we can choose a relatively smaller $k$ to save communication time.

As a comparison, we also prove a theorem for the data-parallel algorithm based on quantized histogram as follows (please refer to the supplementary material for its proof). The theorem basically tells us that the bias introduced by histogram quantization cannot be reduced to zero even if the training data are sufficiently large, and as a result the corresponding algorithm could fail in finding the best attribute.[7] This could be the critical weakness of this algorithm in big data scenario.

**Theorem 4.2** *We denote quantized histogram with $b$ bins of the underlying distribution $P$ as $P^b$, that of the empirical distribution $P_n$ as $P_n^b$, the information gain of $X_j$ calculated under the distribution $P^b$ and $P_n^b$ as $IG_j^b$ and $IG_{n,j}^b$ respectively, and $f_j(b) \triangleq |IG_j - IG_j^b|$. Then, for $\epsilon \leq \min_{j=1,\cdots,d} f_j(b)$, with probability at least $\delta_j(n, f_j(b) - \epsilon))$, we have $|IG_{n,j}^b - IG_j| > \epsilon$.*

## 5 Experiments

In this section, we report the experimental comparisons between PV-Tree and baseline algorithms. We used two data sets, one for learning to rank (LTR) and the other for ad click prediction (CTR)[8] (see Table 1 for details). For LTR, we extracted about 1200 numerical attributes per data sample, and used NDCG [5] as the evaluation measure. For CTR, we extracted about 800 numerical attributes [9], and used AUC as the evaluation measure.

| Table 1: Datasets | | | |
|---|---|---|---|
| Task | #Train | #Test | #Attribute | Source |
| LTR | 11M | 1M | 1200 | Private |
| CTR | 235M | 31M | 800 | KDD Cup |

| Table 2: Convergence time (seconds) | | | | |
|---|---|---|---|---|
| Task | Sequential | Data-Parallel | Attribute-Parallel | PV-Tree |
| LTR | 28690 | 32260 | 14660 | 5825 |
| CTR | 154112 | 9209 | 26928 | 5349 |

According to recent industrial practices, a single decision tree might not be strong enough to learn an effective model for complicated tasks like ranking and click prediction. Therefore, people usually use decision tree based boosting algorithms (e.g., GBDT) to perform tasks. In this paper, we also use GBDT as a platform to examine the efficiency and effectiveness of decision tree parallelization. That is, we used PV-Tree or other baseline algorithms to parallelize the decision tree construction process in each iteration of GBDT, and compare their performance. Our experimental environment is a cluster of servers (each with 12 CPU cores and 32 GB RAM) inter-connected with 1 Gbps Ethernet. For the experiments on LTR, we used 8 machines for parallel training; and for the experiments on CTR, we used 32 machines since the dataset is much larger.

## 5.1 Comparison with Other Parallel Decision Trees

For comparison with PV-Tree, we have implemented an attribute-parallel algorithm, in which a binary vector is used to indicate the split information and exchanged across machines. In addition, we implemented a data-parallel algorithm according to [2, 21], which can communicate both full-grained histograms and quantized histograms. All parallel algorithms and sequential(single machine) version are compared together.

The experimental results can be found in Figure 1a and 1b. From these figures, we have the following observations:

For LTR, since the number of data samples is relatively small, the communication of the split information about the samples does not take too much time. As a result, the attribute-parallel algorithm appears to be efficient. Since most attributes take numerical values in this dataset, the full-grained histogram has quite a lot of bins. Therefore, the data-parallel algorithm which communicates full-grained histogram is quite slow, even slower than the sequential algorithm. When reducing the bins in the histogram to 10%, the data-parallel algorithm becomes much more efficient, however, its convergence point is not good (consistent with our theory – the bias in quantized histograms leads to accuracy drop).

For CTR, attribute-parallel algorithm becomes very slow since the number of data samples is very large. In contrast, many attributes in CTR take binary or discrete values, which make the full-grained histogram have limited number of bins. As a result, the data-parallel algorithm with full-grain histogram is faster than the sequential algorithm. The data-parallel algorithm with quantized histograms is even faster, however, its convergence point is once again not very good.

PV-Tree reaches the best point achieved by sequential algorithm within the shortest time in both LTR and CTR task. For a more quantitative comparison on efficiency, we list the time for each algorithm (8 machines for LTR and 32 machines for CTR) to reach the convergent accuracy of the sequential algorithm in Table 2. From the table, we can see that, for LTR, it costed PV-Tree 5825 seconds, while it costed the data-parallel algorithm (with full-grained histogram[9]) and attribute-parallel algorithm 32260 and 14660 seconds respectively. As compared with the sequential algorithm (which took 28690 seconds to converge), PV-Tree achieves 4.9x speed up on 8 machines. For CTR, it costed PV-Tree 5349 seconds, while it costed the data-parallel algorithm (with full-grained histogram) and attribute-parallel algorithm 9209 and 26928 seconds respectively. As compared with the sequential algorithm (which took 154112 seconds to converge), PV-Tree achieves 28.8x speed up on 32 machines.

We also conducted independent experiments to get a clear comparison of communication cost for different parallel algorithms given some typical big data workload setting. The result is listed in Table 3. We find the cost of attribute-parallel algorithm is relative to the size of training data $N$, and the cost of data-parallel algorithm is relative to the number of attributes $d$. In contrast, the cost of PV-Tree is constant.

Table 3: Comparison of communication cost, train one tree with depth=6.

| Data size | Attribute Palallel | Data Parallel | PV-Tree k=15 |
|---|---|---|---|
| N=1B, d=1200 | 750MB | 424MB | 10MB |
| N=100M, d=1200 | 75MB | 424MB | 10MB |
| N=1B, d=200 | 750MB | 70MB | 10MB |
| N=100M, d=200 | 75MB | 70MB | 10MB |

Table 4: Convergence time and accuracy w.r.t. global voting parameter $k$ for PV-Tree.

| | k=1 | k=5 | k=10 | k=20 | k=40 |
|---|---|---|---|---|---|
| LTR M=4 | 11256/ 0.7905 | 9906/ 0.7909 | 9065/ 0.7909 | 8323/ 0.7909 | 9529/ 0.7909 |
| LTR M=16 | 8211/ 0.7882 | 8131/ 0.7893 | 8496/ 0.7897 | 10320/ 0.7906 | 12529/ 0.7909 |
| CTR M=16 | 9131/ 0.7535 | 9947/ 0.7538 | 9912/ 0.7538 | 10309/ 0.7538 | 10877/ 0.7538 |
| CTR M=128 | 1806/ 0.7533 | 1745/ 0.7536 | 2077/ 0.7537 | 2133/ 0.7537 | 2564/ 0.7538 |

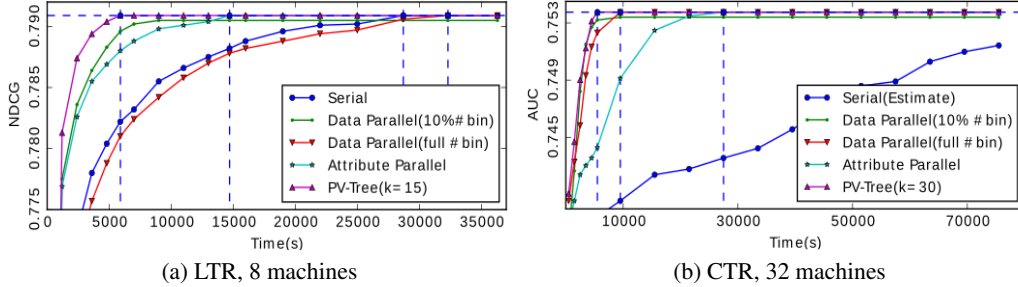

(a) LTR, 8 machines     (b) CTR, 32 machines

Figure 1: Performances of different algorithms

## 5.2 Tradeoff between Speed-up and Accuracy in PV-Tree

In the previous subsection, we have shown that PV-tree is more efficient than other algorithms. Here we make a deep dive into PV-tree to see how its key parameters affect the trade-off between efficiency and accuracy. According to Theorem 4.1, the following two parameters are critical to PV-Tree: the number of machines $M$ and the size of voting $k$.

### 5.2.1 On Different Numbers of Machines

When more machines join the distributed training process, the data throughput will grow larger but the amortized training data on each machine will get smaller. When the data size on each machine becomes too small, there will be no guarantee on the accuracy of the voting procedure, according to our theorem. So it is important to appropriately set the number of machines.

To gain more insights on this, we conducted some additional experiments, whose results are shown in Figure 2a and 2b. From these figures, we can see that for LTR, when the number of machines grows from 2 to 8, the training process is significantly accelerated. However, when the number goes up to 16, the convergence speed is even lower than that of using 8 machines. Similar results can be observed for CTR. These observations are consistent with our theoretical findings. Please note that PV-Tree is designed for the big data scenario. Only when the entire training data are huge (and thus distribution of the training data on each local machine can be similar to that of the entire training data), the full power of PV-Tree can be realized. Otherwise, we need to have a reasonable expectation on the speed-up, and should choose to use a smaller number of machines to parallelize the training.

### 5.2.2 On Different Sizes of Voting

In PV-Tree, we have a parameter $k$, which controls the number of top attributes selected during local and global voting. Intuitively, larger $k$ will increase the probability of finding the globally best attribute from the local candidates, however, it also means higher communication cost. According to our theorem, the choice of $k$ should depend on the size of local training data. If the size of local training data is large, the locally best attributes will be similar to the globally best one. In this case, one can safely choose a small value of $k$. Otherwise, we should choose a relatively larger $k$. To gain more insights on this, we conducted some experiments, whose results are shown in Table 4, where $M$ refers to the number of machines. From the table, we have the following observations. First, for both cases, in order to achieve good accuracy, one does not need to choose a large $k$. When $k \leq 40$, the

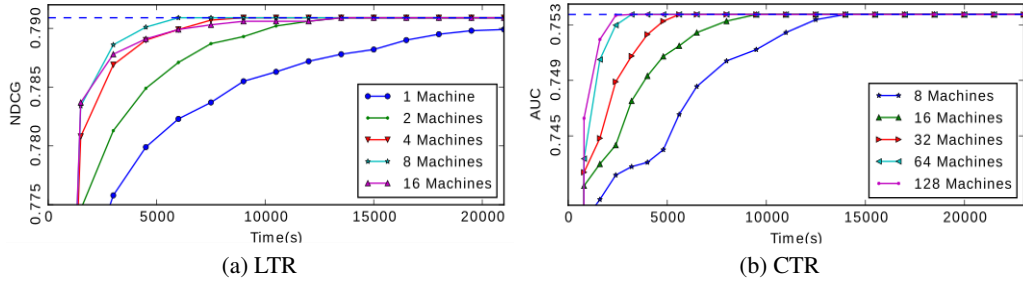

(a) LTR

(b) CTR

Figure 2: PV-Tree on different numbers of machines

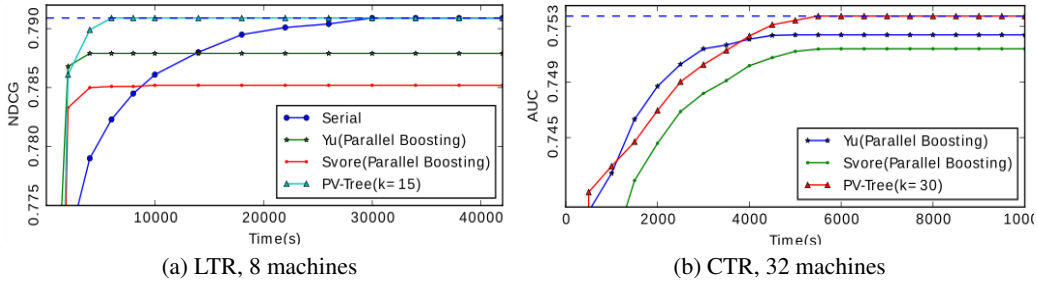

(a) LTR, 8 machines

(b) CTR, 32 machines

Figure 3: Comparison with parallel boosting algorithms

accuracy has been very good. Second, we find that for the cases of using small number of machines, $k$ can be set to an even smaller value, e.g., $k = 5$. This is because, given a fixed-size training data, when using fewer machines, the size of training data per machine will become larger and thus a smaller $k$ can already guarantee the approximation accuracy.

### 5.3 Comparison with Other Parallel GBDT Algorithms

While we mainly focus on how to parallelize the decision tree construction process inside GBDT in the previous subsections, one could also parallelize GBDT in other ways. For example, in [22, 20], each machine learns its own decision tree separately without communication. After that, these decision trees are aggregated by means of winner-takes-all or output ensemble. Although these works are not the focus of our paper, it is still interesting to compare with them.

For this purpose, we implemented both the algorithms proposed in [22] and [20]. For ease of reference, we denote them as *Svore* and *Yu* respectively. Their performances are shown in Figure 3a and 3b. From the figures, we can see that PV-Tree outperforms both *Svore* and *Yu*: although these two algorithms converge at a similar speed to PV-Tree, they have much worse converge points. According to our limited understanding, these two algorithms are lacking solid theoretical guarantee. Since the candidate decision trees are trained separately and independently without necessary information exchange, they may have non-negligible bias, which will lead to accuracy drop at the end. In contrast, we can clearly characterize the theoretical properties of PV-tree, and use it in an appropriate setting so as to avoid observable accuracy drop.

To sum up all the experiments, we can see that with appropriately-set parameters, PV-Tree can achieve a very good trade-off between efficiency and accuracy, and outperforms both other parallel decision tree algorithms designed specifically for GBDT parallelization.

## 6 Conclusions

In this paper, we proposed a novel parallel algorithm for decision tree, called Parallel Voting Decision Tree (PV-Tree), which can achieve high accuracy at a very low communication cost. Experiments on both ranking and ad click prediction indicate that PV-Tree has its advantage over a number of baselines algorithms. As for future work, we plan to generalize the idea of PV-Tree to parallelize other machine learning algorithms. Furthermore, we will open-source PV-Tree algorithm to benefit more researchers and practitioners.

## Footnotes

[3]As indicated by our theoretical analysis and empirical study (see the next sections), a very small $k$ already leads to good performance in PV-Tree algorithm.

[4]Since all analysis are for one arbitrarily fixed node $O$, we omit the notation $O$ here.

[5]In fact, the global voting size can be $\beta k$ with $\beta > 1$. Then the sufficient condition becomes that no less than $[M/\beta + 1]$ of all machines select the most informative attribute.

[6]Please note that using more machines will reduce local computing time, thus the optimal value of machine number may be larger in terms of speed-up.

[7]The theorem for regression holds in the same way, with replacing IG with VG.

[8]We use private data in LTR experiments and data of KDD Cup 2012 track 2 in CTR experiments.

[9]The data-parallel algorithm with 10% bins could not achieve the same accuracy with the sequential algorithm and thus we did not put it in the table.

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
