[Supplementary Material]

# Supplementary Materials for "A Communication-Efficient Parallel Algorithm for Decision Tree"

**Qi Meng**[1*]**, Guolin Ke**[2*]**, Taifeng Wang**[2]**, Wei Chen**[2]**, Qiwei Ye**[2]**,**
**Zhi-Ming Ma**[3]**, Tie-Yan Liu**[2]
[1]Peking University    [2]Microsoft Research
[3]Chinese Academy of Mathematics and Systems Science
[1]qimeng13@pku.edu.cn; [2]{Guolin.Ke, taifengw, wche, qiwye, tie-yan.liu}@microsoft.com;
[3]mazm@amt.ac.cn

This supplementary document is composed of the proofs for Theorem 4.1 (for both regression and classification) and Theorem 4.2 in the paper "A Communication-Efficient Parallel Algorithm for Decision Tree".

First of all, we review the definitions of information gain in classification and variance gain in regression.

**Definition 0.1** *[1][2] In classification, the information gain (IG) for attribute $X_j \in [w_1, w_2]$ at node O, is defined as the entropy reduction of the output $Y$ after splitting node O by attribute $X_j$ at $w$, i.e.,*

$$
\begin{aligned}
IG_j(w; O) &= \mathcal{H}_j - (\mathcal{H}_j^l(w) + \mathcal{H}_j^r(w)) \\
&= P(w_1 \leq X_j \leq w_2)H(Y|w_1 \leq X_j \leq w_2) - P(w_1 \leq X_j < w)H(Y|w_1 \leq X_j < w) \\
&\quad - P(w \leq X_j \leq w_2)H(Y|w \leq X_j \leq w_2),
\end{aligned}
$$

*where $H(\cdot|\cdot)$ denotes the conditional entropy.*

*In regression, the variance gain (VG) for attribute $X_j \in [w_1, w_2]$ at node O, is defined as variance reduction of the output $Y$ after splitting node O by attribute $X_j$ at $w$, i.e.,*

$$
\begin{aligned}
VG_j(w; O) &= \sigma_j - (\sigma_j^l(w) + \sigma_j^r(w)) \\
&= P(w_1 \leq X_j \leq w_2)Var[Y|w_1 \leq X_j \leq w_2] - P(w_1 \leq X_j < w)Var[Y|w_1 \leq X_j < w] \\
&\quad - P(w_2 \geq X_j \geq w)Var[Y|w_2 \geq X_j \geq w],
\end{aligned}
$$

*where $Var[\cdot|\cdot]$ denotes the conditional variance.*

The conditional entropy $H(\cdot|\cdot)$ and the conditional variance $Var(\cdot|\cdot)$ are calculated according to the conditional distribution $P(\cdot|\cdot)$. For $K$ class classification, we assume $Y$ is a discrete random variable which takes value from the set $\{1, \cdots, K\}$ and we have

$$
\begin{aligned}
H(Y|w_1 \leq X_j \leq w_2) &= -\mathbb{E}_{(Y|w_1 \leq X_j \leq w_2)} \log p(Y|w_1 \leq X_j \leq w_2) \quad &(1) \\
&= -\sum_{k=1}^{K} p(Y = k|w_1 \leq X_j \leq w_2) \log p(Y = k|w_1 \leq X_j \leq w_2). \quad &(2)
\end{aligned}
$$

For regression, we assume that $Y$ is a continuous random variable and

$$
\begin{aligned}
Var(Y|w_1 \leq X_j \leq w_2) &= \mathbb{E}\left[(Y - \mathbb{E}[Y|w_1 \leq X_j \leq w_2])^2|w_1 \leq X_j \leq w_2\right] \quad &(3) \\
&= \int p(y|w_1 \leq X_j \leq w_2)y^2 dy - \left(\int p(y|w_1 \leq X_j \leq w_2)y dy\right)^2. \quad &(4)
\end{aligned}
$$

---

[*]Denotes equal contribution. This work was done when the first author was visiting Microsoft Research Asia.

# 1 Theorem 4.1 and its Proof for classification and regression

**Theorem 4.1:** *In classification, suppose we have $M$ local machines, and each one has $n$ training data. PV-Tree at an arbitrary tree node with local voting size $k$ and global majority voting size $2k$ will select the most informative attribute with a probability at least*

$$\sum_{m=[M/2+1]}^{M} C_M^m \left( 1 - \left( \sum_{j=k+1}^{d} \delta_{(j)}(n,k) \right) \right)^m \left( \sum_{j=k+1}^{d} \delta_{(j)}(n,k) \right)^{M-m},$$

*where $\delta_{(j)}(n,k) = \alpha_{(j)}(n) + 4e^{-c_{(j)}n\left(l_{(j)}(k)\right)^2}$ with $\lim_{n\to\infty}\alpha_{(j)}(n) = 0$ and $c_{(j)}$ is constant.*

**Proof for classification:**

Firstly we introduce some notations. We use subscript $n$ to denote the corresponding empirical statistics, which is calculated based on the empirical distribution $\mathbb{P}_n$. Let $w_j^* = argmax_w IG_j(w)$ and $w_{n,j}^* = argmax_w IG_{n,j}(w)$. We denote $IG_j(w_j^*)$ as $IG_j$, which is the largest information gain for attribute $j$. We denote $IG_{n,j}(w_{n,j}^*)$ as $IG_{n,j}$, which is the largest empirical information gain for attribute $j$. As we defined in the main paper, we denote the index of attribute with the $j$-th largest information gain as $(j)$, and its corresponding information gain as $IG_{(j)}$, i.e.,

$$IG_{(1)} \geq \cdots \geq IG_{(j)} \geq \cdots \geq IG_{(d)}.$$

The corresponding empirical information gain for attribute $(j)$ denoted as

$$IG_{n,(1)}, ..., IG_{n,(j)}, ..., IG_{n,(d)}.$$

Note that $IG_{n,(1)}, ..., IG_{n,(j)}, ..., IG_{n,(d)}$ may not be in an increasing order. Similarly, we denote the index of attribute with the $j$-th largest empirical information gain as $(j')$, and its corresponding empirical information gain as $IG_{n,(j')}$,i.e.,

$$IG_{n,(1')} \geq \cdots \geq IG_{n,(j')} \geq \cdots \geq IG_{n,(d')}.$$

Our proof idea is as follows:

*Step 1:* Because $IG_{n,j} \in d(IG_j, l_j(k))$ is a sufficient condition for $(1) \in \{(1'), ..., (k')\}$ to be satisfied[2], we use concentration inequalities to derive a lower bound of probability for $IG_{n,j} \in d(IG_j, l_j(k)), \forall j$, where $d(x,\epsilon)$ denotes the neighborhood of $x$ with radius $\epsilon$.

*Step 2:* By local top-$k$ and global top-$2k$ voting, the most informative attribute $(1)$ will be contained in the global selected set, i.e., $(1) \in \{(1'), ..., (k')\}$, if only no less than $[M/2+1]$ local workers select it. We calculate the probability for the case no less than $[M/2 + 1]$ of all machines select attribute $(1)$ using binomial distribution.

Firstly, we give the probability to ensure $(1) \in \{(1'), ..., (k')\}$. We bound the difference between the information gain and the empirical information gain for an arbitrary attribute. To be clear, we will prove, with probability at least $\delta_j(n,k)$, we have

$$|IG_{n,j} - IG_j| \leq l_j(k).$$

For simplify the notations, let $H_j^l(w) = H(Y|w_1 \leq X_j \leq w)$, $P_j^l(w) = P(w_1 \leq X_j \leq w)$, $H_j^r(w) = H(Y|w \leq X_j \leq w_2)$ and $P_j^r(w) = P(w \leq X_j \leq w_2)$. We decompose $\mathcal{H}_{n,j}^l(w_{n,j}^*) - \mathcal{H}_j^l(w_j^*)$ as

$$\mathcal{H}_{n,j}^l(w_{n,j}^*) - \mathcal{H}_j^l(w_j^*) \tag{5}$$
$$= P_{n,j}^l(w_{n,j}^*)H_{n,j}^l(w_{n,j}^*) - P_j^l(w_j^*)H_j^l(w_j^*) \tag{6}$$
$$= P_{n,j}^l(w_{n,j}^*)H_{n,j}^l(w_{n,j}^*) - P_{n,j}^l(w_j^*)H_j^l(w_j^*) + P_{n,j}^l(w_j^*)H_j^l(w_j^*) - P_j^l(w_j^*)H_j^l(w_j^*). \tag{7}$$

We decompose $\mathcal{H}_{n,j}^r(w_{n,j^*}) - \mathcal{H}_j^r(w_j^*)$ in a similar way, i.e.,

$$\mathcal{H}_{n,j}^r(w_{n,j}^*) - \mathcal{H}_j^r(w_j^*) \tag{8}$$
$$= P_{n,j}^r(w_{n,j}^*)H_{n,j}^r(w_{n,j}^*) - P_{n,j}^r(w_j^*)H_j^l(w_j^*) + P_{n,j}^r(w_j^*)H_j^r(w_j^*) - P_j^r(w_j^*)H_j^r(w_j^*). \tag{9}$$

By adding Ineq.(7) and Ineq.(9), we have the following,

$$P(|IG_{n,j} - IG_j| > l_j(k))$$

$$= P\left(\left|\mathcal{H}_{n,j}^l(w_{n,j}^*) + \mathcal{H}_{n,j}^r(w_{n,j}^*) - (\mathcal{H}_j^l(w_j^*) + \mathcal{H}_j^r(w_j^*))\right| > l_j(k)\right)$$

$$\leq P\left(\left|P_{n,j}^l(w_j^*)H_j^l(w_j^*) - P_j^l(w_j^*)H_j^l(w_j^*)\right| > \frac{l_j(k)}{3}\right) +$$

$$P\left(\left|P_{n,j}^r(w_j^*)H_j^r(w_j^*) - P_j^r(w_j^*)H_j^r(w_j^*)\right| > \frac{l_j(k)}{3}\right) +$$

$$P\left(\left|P_{n,j}^l(w_{n,j}^*)H_{n,j}^l(w_{n,j}^*) - P_{n,j}^l(w_j^*)H_j^l(w_j^*) + P_{n,j}^r(w_{n,j}^*)H_{n,j}^r(w_{n,j}^*) - P_{n,j}^r(w_j^*)H_j^r(w_j^*)\right| > \frac{l_j(k)}{3}\right)$$

$$\triangleq I_1 + I_2 + I_3$$

For term $I_1$, by using Hoeffding's inequality, we have

$$I_1 \leq P\left(H_j^l(w_j^*) \times \left|P_j^l(w_j^*) - P_{n,j}^l(w_j^*)\right| > \frac{l_j(k)}{3}\right) \tag{10}$$

$$\leq P\left(\left|P_j^l(w_j^*) - P_{n,j}^l(w_j^*)\right| > \frac{l_j(k)}{3H_j^l(w_j^*)}\right) \tag{11}$$

$$\leq 2\exp\left(-\frac{2nl_j(k)^2}{9(H_j^l(w_j^*))^2}\right) \tag{12}$$

Similarly, for term $I_2$, we have

$$I_2 \leq 2\exp\left(-\frac{2nl_j(k)^2}{9(H_j^r(w_j^*))^2}\right) \tag{13}$$

Let $c_j = \min\left\{\frac{2}{9(H_j^l(w_j^*))^2}, \frac{2}{9(H_j^l(w_j^*))^2}\right\}$, we have

$$I_1 + I_2 \leq 4\exp\left(-c_j nl_j(k)^2\right). \tag{14}$$

For the term $I_3$, we have

$$J$$
$$= P_{n,j}^l(w_{n,j}^*)H_{n,j}^l(w_{n,j}^*) - P_{n,j}^l(w_j^*)H_j^l(w_j^*) + P_{n,j}^r(w_{n,j}^*)H_{n,j}^r(w_{n,j}^*) - P_{n,j}^r(w_j^*)H_j^r(w_j^*)$$
$$= \frac{1}{n}\sum_{i=1}^n I(w_1 \leq x_{i,j} \leq w_{n,j}^*)H_{n,j}^l(w_{n,j}^*) + \frac{1}{n}\sum_{i=1}^n I(w_{n,j}^* < x_{i,j} \leq w_2)H_{n,j}^r(w_{n,j}^*)$$
$$- \frac{1}{n}\sum_{i=1}^n I(w_1 \leq x_{i,j} \leq w_j^*)H_j^l(w_j^*) - \frac{1}{n}\sum_{i=1}^n I(w_j^* < x_{i,j} \leq w_2)H_j^r(w_j^*),$$

where $x_{i,j}$ is the $j$-th attribute for the $i$-th instance in the training set.

Let $\Theta$ denote the set of all possible values of $(p_1^l, p_1^r, \cdots, p_{K-1}^l, p_{K-1}^r, w_j)$, where $p_k^l = P(Y = k|w_1 \leq X_j \leq w_j)$ and $p_k^r = P(Y = k|w_j < X_j \leq w_2)$. Define the criterion function $\mathbb{M}(\theta) = Pm_\theta$, where $m_\theta(x, y) = -\log p_k^l I(w_1 \leq x \leq w_j) - \log p_k^r I(w_2 \geq x > w_j)$ if $y = k$. The vector $\theta^* = (p_1^{l*}, p_1^{u*}, \cdots, p_{K-1}^{l*}, p_{K-1}^{u*}, w_j^*)$ maximizes $\mathbb{M}(\theta)$, while $\theta_n^* = (p_{n,1}^{l*}, p_{n,1}^{r*}, \cdots, p_{n,K-1}^{l*}, p_{n,K-1}^{r*}, w_{n,j}^*)$ minimizes $\mathbb{M}_n(\theta)$. Straightforward algebra shows that

$$(m_\theta - m_{\theta^*})(X, Y) = I(Y = k)[(\log p_k^{l*} - \log p_k^{r*})(I(w_1 \leq X \leq w_{j,n}^*) - I(w_1 \leq X < d_j^*)) \tag{15}$$
$$+ (\log p_{n,k}^{l*} - \log p_k^{l*})I(w_1 \leq X \leq w_{n,j}^*) \tag{16}$$
$$+ (\log p_{n,k}^{u*} - \log p_k^{r*})I(w_{n,j}^* \leq X \leq w_2)] \tag{17}$$

By following the proof of Theorem 1 in [3], we can get that $n^{2/3}I_3$ converges to $c_2 \max_t Q(t)$, where $c_2$ is a constant and $Q(t)$ is composed by the standard two-sided Brownian Motion [3]. Therefore, we have

$$P\left(|J| > c_2 n^{-\frac{2}{3}} q_\alpha\right) < \alpha. \tag{18}$$

where $q_\alpha$ is the upper $\alpha$-quantile of $\max_t Q(t)$. Let $c_2 n^{-\frac{2}{3}} q_{\alpha_j(n)} = \frac{l_j(k)}{3}$. With probability at most $\alpha_j(n)$, we have $IG_{n,j}(w_j^*) - IG_{n,j} > \frac{l_j(k)}{2}$, i.e.,

$$I_2 = P\left(|J| > \frac{l_j(k)}{3}\right) < \alpha_j(n) \tag{19}$$

By combining Inequalities (14) and (19), we have, with probability at most $\delta_j(n,k) = \alpha_j(n) + 4\exp\left(-c_j n l_j(k)^2\right)$,

$$|IG_{n,j} - IG_j| > l_j(k). \tag{20}$$

Thus we can get

$$P\left(\left|IG_{n,(j)} - IG_{(j)}\right| \le l_j(k), \forall j \ge k+1\right) \ge 1 - \sum_{j=k+1}^{d} \delta_{(j)}(n,k). \tag{21}$$

By binomial distribution, we can derive the results in the theorem. $\square$

**Proof for regression:**

The proof is similar to classification. We continue to use notations in the previous section and just substitute $IG$ to $VG$.

Similarly, we will prove, with probability at least $\delta_j(n,k)$, we have

$$|VG_{n,j} - VG_j| \le l_j(k).$$

By the definition of variance gain, we have the following,

$$
\begin{aligned}
&P\left(|VG_{n,j} - VG_j| > l_j(k)\right) \\
&\le P(|\sigma_{n,j}^l(w_{n,j}^*) + \sigma_{n,j}^r(w_{n,j}^*) - \sigma_j^l(w_j^*) - \sigma_j^r(w_j^*)| > l_j(k)) \\
&\le P\left(\left|P_{n,j}^l(w_j^*)\sigma_j^l(w_j^*) - P_j^l(w_j^*)\sigma_j^l(w_j^*)\right| > \frac{l_j(k)}{3}\right) + \\
&\quad P\left(\left|P_{n,j}^r(w_j^*)\sigma_j^r(w_j^*) - P_j^r(w_j^*)\sigma_j^r(w_j^*)\right| > \frac{l_j(k)}{3}\right) + \\
&\quad P\left(\left|P_{n,j}^l(w_{n,j}^*)\sigma_{n,j}^l(w_{n,j}^*) - P_{n,j}^l(w_j^*)\sigma_j^l(w_j^*) + P_{n,j}^r(w_{n,j}^*)\sigma_{n,j}^r(w_{n,j}^*) - P_{n,j}^r(w_j^*)\sigma_j^r(w_j^*)\right| > \frac{l_j(k)}{3}\right) \\
&\triangleq I_1 + I_2 + I_3
\end{aligned}
$$

For term $I_1$, by using Hoeffding's inequality, we have

$$I_1 \le P\left(\sigma_j^l(w_j^*) \times \left|P_j^l(w_j^*) - P_{n,j}^l(w_j^*)\right| > \frac{l_j(k)}{3}\right)$$

$$\le P\left(\left|P_j^l(w_j^*) - P_{n,j}^l(w_j^*)\right| > \frac{l_j(k)}{3\sigma_j^l(w_j^*)}\right) \tag{22}$$

$$\le 2\exp\left(-\frac{2nl_j(k)^2}{9(\sigma_j^l(w_j^*))^2}\right) \tag{23}$$

Similarly, for term $I_2$, we have

$$I_2 \le 2\exp\left(-\frac{2nl_j(k)^2}{9(\sigma_j^r(w_j^*))^2}\right) \tag{24}$$

Let $c_j = \min\left\{\frac{2}{9(\sigma_j^r(w_j^*))^2}, \frac{2}{9(\sigma_j^l(w_j^*))^2}\right\}$, we have

$$I_1 + I_2 \le 4\exp\left(-c_j n l_j(k)^2\right). \tag{25}$$

For the term $I_3$, let $J = P_{n,j}^l(w_{n,j}^*)\sigma_{n,j}^l(w_{n,j}^*) - P_{n,j}^l(w_j^*)\sigma_j^l(w_j^*) + P_{n,j}^r(w_{n,j}^*)\sigma_{n,j}^r(w_{n,j}^*) - P_{n,j}^r(w_j^*)\sigma_j^r(w_j^*)$. According to Theorem 2.2 established by [3], the following holds,

$$P\left(|J| > c_2 n^{-\frac{2}{3}} q_\alpha\right) < \alpha. \tag{26}$$

where $c_2$ is a constant for fixed distribution $P$ and $q_\alpha$ is the upper $\alpha$-quantile of the standard two-sided Brownian Motion [3]. With probability at most $\alpha_j(n)$, we have $|J| > \frac{l_j(k)}{3}$, i.e.,

$$I_3 = P\left(|J| > \frac{l_j(k)}{3}\right) < \alpha_j(n) \tag{27}$$

By combining Ineq.(25) and (27), we have, with probability at most $\delta_j(n,k) = \alpha_j(n) + 4\exp\left(-c_j n l_j(k)^2\right)$,

$$|VG_{n,j} - VG_j| > l_j(k). \tag{28}$$

Thus we can get

$$P\left(\left|VG_{n,(j)} - VG_{(j)}\right| \le h, \forall j \ge k+1\right) \ge 1 - \sum_{j=k+1}^{d} \delta_{(j)}(n,k). \tag{29}$$

By binomial distribution, we can derive the results in the theorem. $\square$

## 2 Theorem 4.2 and its proof

**Theorem 4.2:** *We denote quantized histogram with $b$ bins of the underlying distribution $P$ as $P^b$, that of the empirical distribution $P_n$ as $P_n^b$, the information gain of $X_j$ calculated under the distribution $P^b$ and $P_n^b$ as $IG_j^b$ and $IG_{n,j}^b$ respectively, and $f_j(b) \triangleq |IG_j - IG_j^b|$. Then, for $\epsilon \le \min_{j=1,\cdots,d} f_j(b)$, with probability at least $\delta_j(n, f_j(b) - \epsilon))$, we have $|IG_{n,j}^b - IG_j| > \epsilon$.*

**Proof:**
First, $|IG_{n,j}^b - IG_j| = |IG_{n,j}^b - IG_j^b + IG_j^b - IG_j| \ge \left||IG_{n,j}^b - IG_j^b| - |f(b)|\right|$. Second, when $n$ is large enough, we have $|f(b)| - |IG_{n,j}^b - IG_j^b| > \epsilon$ with probability $\delta_j(n, f_j(b) - \epsilon))$ for $\epsilon \le \min_{j=1,\cdots,d} f_j(b)$. Thus, the proposition is proven. $\square$

## Footnotes

[2]In order to $(1) \in \{(1'), ..., (k')\}$, the number of $IG_{n,j}$ which is larger than $IG_{n,(1)}$ is at most $k-1$.