[Reviews · NeurIPS 2016]

Reviewer 1

Summary

Authors propose a data-parallel algorithm for learning decision trees, which greatly improves communication efficiency compared to previously proposed algorithms. Instead of computing the global histogram for all attributes, each local worker votes to k attributes, and global histogram is computed only for top 2k attributes that receive most votes. Authors provide a theoretical analysis that characterizes the probability an optimal attribute would be chosen. Empirical comparisons against previously proposed attribute-parallel and data-parallel algorithms are provided and results are encouraging.

Qualitative Assessment

The proposed voting scheme is an intuitive and appealing way of reducing the communication cost: first figure out which attributes are important only with crude information (votes instead of histograms), and then concentrate the communication cost on attractive set of attributes. The guarantee in the Theorem 4.1 seems to be quite week, as the success probability would exponentially decrease as a function of d. However, empirical results alleviate the concern, as near-optimal performance seem to be achieved even when there are 128 machines and only one attributes are elected (k=1). Minor comments: 1. Line 38 in the appendix is a bit hand-wavy; would be nice if the proof of equation (7) could be included in the appendix. 2. 'attribute (1)' and 'h_{(j)}(k)' in line 156 do not seem to be defined. I guess authors mean the most informative attribute by '(1)'.

Confidence in this Review

2-Confident (read it all; understood it all reasonably well)


Reviewer 2

Summary

This paper proposes a new parallel algorithm for training decision trees (for classification or regression). The main idea of this algorithm is to distribute the data on different nodes and then at each node, to identify the top-k splits on each data subset, determine globally to top-2k splits by voting individual top-k lists and then identify the best ones among them by combining histograms from individual workers. A theoretical analysis is performed that gives a lower bound on the probability to find the globally optimal split depending on the main method parameters. Several experiments are conducted on two large-scale datasets, where PV-tree are shown to outperform other parallel decision tree implementation.

Qualitative Assessment

Given the popularity of decision trees, proposing an efficient parallel implementation of this method is of course very relevant. The proposed parallelization is original with respect to existing methods and it should indeed lead to less communications than other methods. The theoretical analysis is sound and I like the discussion of the impact of the main problem and method parameters that follows from the lower bound provided in theorem 4.1. Experiments are conducted on two very large problems, where, in the limit of the tested settings (see below), PV-tree is clearly shown to outperform other parallel implementations, in terms of both computing times to reach a given accuracy level and communication costs. I nevertheless have two major concerns with the proposed parallelization. First, given the way it works, the performance of the algorithm is clearly affected by the number of workers and this number thus needs to be tuned for optimal performance. This is a serious drawback as It means that the algorithm can not benefit from large number of workers. From my understanding, this problem is not shared by other parallelizations. Using more machines with these methods might not always give significant improvement in terms of computing times but at least, it will not deteriorate predictive performance. While the experiments show the impact of the number of workers on PV-trees, the impact on other methods should be also studied in comparison. Experiments should show if PV-tree is still the best method on LTR (resp. CTR) when the number of machines grows (far) beyond 8 (resp. 32) machines. Second, I'm also puzzled by this statement from the authors in Section 5.2.1 (and similar statements are given in Section 4): "Only when the entire training data are huge (and thus distribution of the training data on each local machine can be similar to that of the entire training data), the full power of PV-Tree can be realized." If this is indeed true, then I'm wondering why it's actually needed to distribute the data over different machines and why just selecting the best split over a single random subset of N/M examples at each node is not enough. (Or similarly is the tree built using PV-tree really better than a tree grown using only the subset of data available at one of the worker?) Indeed, if the distribution of the training data on each local machine is assumed, for the method to work, to be similar to that of the entire data, then the local top-k splits are expected to be the same as the global top-k splits and there is no need to vote over several workers. I think that this should be checked in the experiments. Do we really need to vote over several workers or is one worker not actually enough, in the conditions (in terms of the number of workers) where PV-tree works best? The paper is very pleasant to read. Several important details are however missing in the textual description of the algorithms and in the pseudocodes. Some points that need to be clarified are as follows: - How are PV-tree and the competitors implemented? (language, framework, etc) - The pseudocode does not clearly indicate how and when the data is distributed. At each node or only once globally? What about other methods? What is included in their communication cost? - I don’t find the pseudo-codes very clear, probably because of the use of an object oriented formalism. Most functions/methods are not defined and it’s not always easy to guess what they are doing from their names. I think it could be improved. - Which information is contained in the histograms, in particular in the case of continuous attributes? How is this information merged between workers to yield the global histogram and how is the split threshold determined from this histogram? I don't have a clear idea of what kind of information is actually communicated between the machines. - Is the global split found using the call ‘globalHistograms.FindBestSplit()’ at the end of Algorithm 3 exactly the same as the split that would have been obtained with all available data (assuming the optimal variable is among the 2k selected ones).

Confidence in this Review

2-Confident (read it all; understood it all reasonably well)


Reviewer 3

Summary

In this paper, a new data-parallel algorithm is proposed for the parallel training of decision trees. This algorithm is declared to have achieved a better trade-off between accuracy and efficiency, compared to existing parallel decision tree algorithms. Specifically, both local voting and global voting are performed before the most informative attribute is identified from the collection of local machines. The proposed algorithm sacrifices computational efficiency in local machines for the reduction of overall communication cost.

Qualitative Assessment

The presentation is clear, though the reference seems not quite up-to-date. (No citation published later than 2012.) The experiments and results sustained the claim that PV-Tree outperforms existing parallel decision tree algorithms, if not by a significant margin. Specifically, Table 2 and Table 3 illustrate the advantage of PV-Tree over other algorithms in the aspect of efficiency and communication cost, respectively. Since it is declared in the abstract that PV-Tree outperforms other algorithms in the trade-off between accuracy and efficiency, a question may be raised, which is, how is the evaluation of the trade-off defined? Is there any weight involved in the balance between accuracy and efficiency, or, are they equally important? This seems not have been illustrated. Results are pretty well analyzed. For instance, the limitation of PV-Tree has been illustrated that it is designed for the big data scenario, and that larger number of machines does not necessarily render faster training than a smaller number of machines, when the data split in each machine is small. The tables are good, but one suggestion might be helpful: An explanatory figure may better visualize the algorithm to be proposed.

Confidence in this Review

1-Less confident (might not have understood significant parts)


Reviewer 4

Summary

This paper introduces a new parallel algorithm for decision tree, called PV-Tree. The main goal is to significantly decrease the communciation cost of such decision tree building. To do that, the authors propose to be non-comprehensive in each machine (which run in paralle) when communicating information about the splits. The auhtors prove a theorem to justify their approach. Indeed with high probability PV-tree manage to find the actual best split (i.e. the one which would be find if all data were processed in one single machine). Finally, experiments are presented on two relatively large datasets, and comparisons with other parallel decision trees are made.

Qualitative Assessment

The subject of the paper is interesting, especially in the context of big data, where obviously standard implementation of decision trees are not reasonable (or even possible). The paper is well written and clear. My main concerns are about the remarks following theorem 4.1: 1. size of local training data n: it is said that if n is large enough, PV-tree will select the best attribute with almost probability one. I agree that the probability is increasing with n, but I wonder why the question on computation time is not addressed here. My point applies to the sentence l. 126 where authors say that PV-tree can scale independently of the number of samples. 2. input dimension d: the conclustion of the remark is that the bound in the theorem is not sensitive with d. But the rest of the remark suggests the opposite. How do we know that the \delta_{(j)} are small enough to compensate the sum (which gets 100 times more temrs) ? 3. The same here, it is said that a probability tends to one as M increases, but the rated of convergence is important here. 4. I think there is a mistake here: "as k increases, the lower bound decreases". According to me, the more k increase, the more chances we have to get the actual best attribute in each machine, hence the lower bound should increase. My last main concern is about the sentence l.247-249. How do we know that the training data in each local machine is similar to that of the entire data ? I think this point is central in the big data context, and the authors never address the fact that the way the data are distributed in several machines is very important. Indeed, it is not because every machine sees a huge amount of data that every samples have the same distribution. Minor: - l. 145: it's esay to PROVE - l. 230-231: I think #data and #attribute should be avoided.

Confidence in this Review

2-Confident (read it all; understood it all reasonably well)


Reviewer 5

Summary

The paper introduces a novel algorithm, named the Parallel Voting Decision Tree (PV-Tree), which effectively parallelizes the training process of the decision tree. The algorithm partitions the training data onto the different machines and then proceeds in two steps. In the first step, the algorithm finds the top-k attributes from the local data for each machine. In the second step, the algorithm globally finds the top 2k attributes among the previous set of local attributes. Finally, the full-grained histograms of these top 2k attributes are collected from the local machines in order to find their global distributions and identify the best attribute and its split point. The algorithm is much more efficient than existing algorithms that parallelize the training process of decision trees. It is more efficient than both data-parallel and attribute-parallel algorithms due to a very low communication cost. The algorithm only communicates the indices of the top k attributes for every machine and the histograms of the globally top 2k attributes. The author also proves the best attribute is accurately chosen by the algorithm with a very large probability that converges to one as the training sample size increases. The accuracy holds regardless of the value of k chosen.

Qualitative Assessment

The proposed novel algorithm is very well supported with theoretical results as well as experimental results. Overall a very good paper. There were a few issues though that were not clearly explained. 1. The global voting size through out the paper has been chosen to be 2k, which is twice the local voting size. The author mentions that it can be chosen to be anything greater than the local voting size. But it has not been discussed how the global voting size effects the efficiency or accuracy of the algorithm. It has also not been explained why 2k was chosen. 2. Lines 153-158: It is not very clear how the input dimension, d affects the bound. The author mentions that the lower bound decreases with increasing dimension, but concludes by saying that the bound is not sensitive to the dimension. The explanation given is not clear. 3. Lines 247-249: The author mentions that the algorithm works well for large data sets. How large does a training set need to be for the algorithm to work well? 4. Figure 2(b): The convergence rate for 64 machines seems to be lower in the figure than both 32 machines and 128 machines. The accuracy appears to be the same regardless of the number of machines. Is there a reason why the algorithm works worse when 64 machines are used? Some minor points: Line 88, In the equation, H(Y|.) has not been defined earlier. Line 98, "However its design principal..." should be "However its principal design..." . Line 156: h(j)(k) has not been introduced before. It is probably a typo and should be l(j)(k) instead.

Confidence in this Review

2-Confident (read it all; understood it all reasonably well)


Reviewer 6

Summary

This paper proposes a communication-efficient algorithm for decision trees when data are split across a number of machines. The PV-Tree method adopts both local and global voting mechanism to leverage local information to reduce the communication cost. Factors that affect the statistical performance are discussed based on the theorem and numerical experiments further support the analysis.

Qualitative Assessment

The idea of leveraging local information to avoid redundant communication is novel. The main communication cost of the algorithm is on the best attribute identification. It seems to me that the reason for the choice of 2k is that as long as [M/2+1] of all machines select an attribute, it is guaranteed to be in the global list. As mentioned in the footnote, we can choose other multiples of k, so I wonder if the performance would be significantly affected if more or less candidates are considered. Now we treat the local candidates equally when aggregating to the global list. Would assigning different weights a reasonable idea to improve the selection? The numerical experiments are comprehensive and well support the algorithm and theoretical results. It would be better to give a definition or some explanation of the NDCG measure. In general, the algorithm is novel and can have many applications. The paper is well written and organized.

Confidence in this Review

2-Confident (read it all; understood it all reasonably well)